# Dimension Prediction and Microstructure Study of Wire Arc Additive Manufactured 316L Stainless Steel Based on Artificial Neural Network and Finite Element Simulation

**DOI:** 10.3390/mi15050615

**Published:** 2024-04-30

**Authors:** Yanyan Di, Zhizhen Zheng, Shengyong Pang, Jianjun Li, Yang Zhong

**Affiliations:** State Key Laboratory of Materials Processing and Die & Mould Technology, Huazhong University of Science and Technology, Wuhan 430074, Chinajianjun@hust.edu.cn (J.L.);

**Keywords:** wire arc additive manufacturing, artificial neural network, finite element, dimension, heat transfer, microstructure

## Abstract

The dimensional accuracy and microstructure affect the service performance of parts fabricated by wire arc additive manufacturing (WAAM). Regulating the geometry and microstructure of such parts presents a challenge. The coupling method of an artificial neural network and finite element (FE) is proposed in this research for this purpose. Back-propagating neural networks (BPNN) based on optimization algorithms were established to predict the bead width (BW) and height (BH) of the deposited layers. Then, the bead geometry was modeled based on the predicted dimension, and 3D FE heat transfer simulation was performed to investigate the evolution of temperature and microstructure. The results showed that the errors in BW and BH were less than 6%, and the beetle antenna search BPNN model had the highest prediction accuracy compared to the other models. The simulated melt pool error was less than 5% with the experimental results. The decrease in the ratio of the temperature gradient and solidification rate induced the transition of solidified grains from cellular crystals to columnar dendrites and then to equiaxed dendrites. Accelerating the cooling rate increased the primary dendrite arm spacing and δ-ferrite content. These results indicate that the coupling model provides a pathway for regulating the dimensions and microstructures of manufactured parts.

## 1. Introduction

Wire arc additive manufacturing (WAAM) is an advanced direct energy deposition method that mainly includes gas tungsten arc welding (GTAW), plasma arc welding (PAW), and gas metal arc welding (GMAW) [1]. This technology uses one or more arcs as a heat source to fabricate components with complicated shapes by depositing the material layer by layer in the absence of a mold [2,3]. Therefore, WAAM is highly efficient [4], low-cost [5,6], and flexible [7] in the forming process, making it promising for the production of big and complex structural components in certain industries such as aviation, aerospace and nuclear power [8,9]. However, due to its complex physical processes and variable process parameters, controlling the dimension accuracy and microstructure of manufactured products remains a significant challenge in WAAM processes.

Some scholars have concentrated on studying the effect of WAAM processes on the dimension, microstructure, and performance of parts, such as deformation [10], cracks [11], microstructure components [12,13], and the mechanical properties [14,15] of the formed parts. These studies provide some guidance for understanding and applying WAAM technology. However, the dimensions and microstructure of the deposited layers are influenced by a combination of factors. Different materials have different thermal properties, and different process parameters result in different temperature distributions and flow within the molten pool, which further affect the pool morphology and solidification behavior, leading to changes in the dimensions and microstructure of the deposited layers. Addressing these influencing factors inevitably involves a series of experiments, adjustments, and optimizations during the additive manufacturing process, consuming a significant amount of time and resources.

It is well known that numerical computational methods are considered an appropriate alternative, including artificial neural network (ANN), finite element (FE) simulation, etc. Cho et al. [16] and Fu et al. [17] adopted the response surface model to realize the prediction of WAAM-ed bead geometry. Mu et al. [18] invented an adaptive controller with an MPC algorithm and linear autoregressive model to enhance the geometrical precision of the CMT process. Xiong et al. [19] compared the performance of an artificial neural network and second-order regression method in forecasting the height (BH) and width (BW) of a bead. According to the work of Ikeuchi et al. [20], ANN modeling may be more suitable for the prediction of profile additives manufactured using cold spray than Gaussian modeling. Xian et al. [21] and Hejripour et al. [22], respectively, revealed the effects of the cooling rate (CR) on the α content of a WAAM-ed Ti-6Al-4V alloy and the austenite content of WAAM-ed duplex stainless steel through FE thermal simulation. The prediction of the relationship between the thermal history and solid-state phases was achieved using the FE method by Mishra et al. [23]. A high CR caused the martensitic phase, while a low CR caused a predominance of bainite and ferrite. Others investigated the effect of forming parameters on grain growth through more complex multi-scale models [24,25,26,27]. Clearly, numerical simulation has provided significant advantages in addressing issues within the WAAM process. However, it also presents some limitations and challenges, such as complex models describing physical processes and the ideal cuboid geometry used in most FE simulations to represent sedimentary layers [28,29]. Therefore, there is a need for an efficient, intelligent, and less complex method to study the influence of process parameters on the dimensions and microstructure of manufactured parts, thereby achieving process design and optimization.

In this study, a geometry dimension prediction-heat transfer model was developed to investigate the evolutionary behavior of the geometry, temperature, and microstructure of a deposited layer. The prediction of the geometrical dimensions under different processes was achieved by ANN, and the influence of the temperature evolution on the microstructure under different deposition parameters was analyzed by an FE heat transfer model. This study provides a way of achieving the prediction of the dimensions and microstructure of a deposited layer fabricated via the WAAM process.

## 2. Experiment

The substrate and welding wire used for the experiments were SS316L wires. The SS316L wire is a flux-cored wire with a diameter of 1.2 mm; the chemical composition of the welding wire is shown in Table 1. The WAAM system was mainly composed of a Fronius CMT 4000 welding machine, KUKA KR 30 HA six-axis industrial robot, worktable, and protective gas device. Before the experiment, the substrate was polished with a grinding machine for clearing the oxidized surface. The shielding gas was Ar and CO_2_, with a volume ratio of 8:2. It is worth noting that CMT equipment has same average voltage (AU) and average current (AI) at the same WFS, even though the transient voltage (TU) and transient current (TI) change slightly during deposition. The WAAM parameters are shown in Table 2.

The single-track deposition layer geometry morphology is shown in Figure 1. To get the width and height more accurately, the top view and cross section’s contours were obtained by performing an edge detection process in MATLAB. Additionally, when obtaining the geometry data, BW and BH were measured at three positions of the deposition layer (see Figure 1). Finally, the average values were calculated to obtain the corresponding BW and BH for the process parameter (Appendix A). The microstructure of the specimens was observed using an ultra-deep 3D stereomicroscope (VHX-1000C, Keyence Co., Osaka, Japan) after being cut, ground, polished and etched. X-ray diffraction (XRD) was chosen for phase recognition and a Wilson 430SVD micro-hardness tester with a load of 4.9 kg was used for Vickers micro-hardness testing.

## 3. Numerical Model

The computational process of the model is mainly composed of 3 parts (see Figure 2). First, to get closer to the real geometry of the deposition layer, we established an ANN model to predict the BH and BW under different process parameters and determined a more suitable prediction model by comparing the prediction results of the BPNN, GA-BPNN, PSO-BPNN, and BAS-BPNN models. Second, the predicted BWs and BHs were brought into the ellipse function (Equation (1) [30]), and the function was solved in MATLAB R2019a to obtain a geometric model close to the real one. Finally, a FE heat transfer model was established and solved computationally based on the geometric model and verified with experiments. By extracting the solidification characteristic parameters (temperature gradient and solidification rate), the intrinsic connection between temperature changes and the microstructure was investigated to achieve the qualitative prediction of the microstructure. Detailed information is described in the following sections.
(1)$$\frac{x^{2}}{{\left(BW/2\right)}^{2}}+\frac{y^{2}}{BH^{2}}=1$$

### 3.1. Dimension Prediction Model

BP artificial neural networks are a very mature method in regression analysis. They usually consist of an input, hidden, and output layer (I-H-O, Figure 3) [31]. In this paper, the WAAM process parameters were the input neurons and the BW and BH of the deposition layer were the output neurons. During the forward propagation, the input sampled signals were transmitted in the order of the input, hidden, and output layer, and after a nonlinear transformation, the output signal was generated and the actual output was compared with the predicted result. When the difference was large, backward propagation processing of the error was carried out, back propagating layer by layer through the H–I layer and spreading the error to all units, using the error signal obtained from each layer as the basis for adjusting the weight value (W) of each unit. Through successive iterations, the inter-layer parameters were updated, the W and bias values (B) corresponding to the smallest errors in width and height were identified, and finally, the prediction was achieved.

However, BPNN learns and converges slowly and is prone to local minima. To improve this situation, we used an intelligent algorithm to get the global optimum W and B.

A genetic algorithm (GA) is a process of selection, mutation, and crossover from one generation to the next, through which individuals more adapted to the environment are left behind [32,33]. Particle swarm optimization algorithm (PSO) is a reference to the behavior of birds flying to the best food every time when searching for food randomly [34]. The beetle antenna search algorithm (BAS) is the process by which the beetle uses its left and right antennae to randomly explore towards food that has a higher concentration of odor [35].

These algorithms have a global search feature that avoids the local optimum. Therefore, we can improve the W and B of the BPNN model by using the “individuals” and “food”. The computational flow of three algorithms is shown in Figure 4, Figure 5 and Figure 6. During prediction, in GA-BPNN, the crossover probability=0.86 and the genetic probability=0.2095. In PSO-BPNN, the inertia weight=0.75, the individual learning factor=2, and the social learning factor=2. In BAS-BPNN, the decay coefficient=0.6 and the antennae spacing=4.

### 3.2. FE Heat Transfer Model

#### 3.2.1. 3D Geometric Model

The ellipse function model can effectively describe the cross-section profile of a single-track layer in the CMT process [31]. Therefore, based on the predicted width and height, the elliptic geometric model of a single track was established for the FE thermal simulation. Furthermore, to improve the calculation accuracy and efficiency, we adopted a local encrypted grid in the calculation process, as shown in Figure 7.

#### 3.2.2. Heat Transfer Control Equation

The 3D transient heat conduction equation was employed as the governing equation to calculate the temperature change during the WAAM process, and can be described mathematically as [36]:(2)$$\rho c\frac{\partial T}{\partial t}=\frac{\partial }{\partial x}\left(\lambda \frac{\partial T}{\partial x}\right)+\frac{\partial }{\partial y}\left(\lambda \frac{\partial T}{\partial y}\right)+\frac{\partial }{\partial z}\left(\lambda \frac{\partial T}{\partial z}\right)+\eta UI$$
where c stands for the specific heat capacity, *I* represents the WAAM current, λ denotes the thermal conductivity of the SS316L. U is the WAAM voltage, *t* indicates the heat transfer time, and η is the thermal efficiency coefficient—the value for this experiment was set to 0.7. ρ is the material density and *T* denotes the temperature. Furthermore, the physical properties of the SS316L used in the simulation (see Figure 8) were calculated in JMatPro 8.0 software.

The energy input was determined using a double ellipsoidal heat source. According to this model, the volumetric heat density distribution that occurred in the front ellipsoid and remaining part can, respectively, be determined as follows:(3)$$q_{1}\left(x,y,z\right)=\frac{6\sqrt{3}f_{1}\eta UI}{a_{1}bc\pi \sqrt{\pi }}exp\left(-\frac{3x^{2}}{a_{1}^{2}}-\frac{3y^{2}}{b^{2}}-\frac{3z^{2}}{h^{2}}\right)$$
(4)$$q_{2}\left(x,y,z\right)=\frac{6\sqrt{3}f_{2}Q}{a_{2}bc\pi \sqrt{\pi }}exp\left(-\frac{3x^{2}}{a_{2}^{2}}-\frac{3y^{2}}{b^{2}}-\frac{3z^{2}}{h^{2}}\right)$$
where $a_{1}$ denotes the length of the front ellipsoid semi-axis, $a_{2}$ represents the length of the posterior ellipsoidal semiaxis, *b* is the heat source width and *h* denotes the depth.

The initial boundary condition is:(5)$$T_{0}(x,y,z)=T_{e}$$

The boundary conditions during the calculation are:(6)$$\lambda \frac{\partial T}{\partial x}n_{x}+\lambda \frac{\partial T}{\partial y}n_{y}+\lambda \frac{\partial T}{\partial y}n_{y}=\beta \left(T_{e}-T\right)+\varepsilon C\left(T_{e}^{4}-T^{4}\right)$$
where $T_{e}$ is the room temperature, *β* is the convective heat exchange system, ε denotes the surface emissivity, and C represents the Stefan–Boltzmann constant.

## 4. Results and Discussion

### 4.1. Geometry Dimension

The actual and predicted values of the BW and BH are shown in Table 3 and Table 4, and the errors are shown in Figure 9. The prediction errors for the width ranged from 0.00943% to 7.1076% and for height from 0.0011% to 10.0595%, which indicates that the established model was able to achieve the prediction of the deposited layer dimension well. In addition, from the prediction results, there were some differences between the predictions of different models. When predicting the BW, the BPNN model had the largest absolute error among the four models, which was 7.1076% (WAAM parameter 2), and the smallest absolute error value was also generated by the BPNN model, which was only 0.0094%. A similar situation was observed for the prediction of the BH (WAAM parameters 1 and 8). This could be the result of the BPNN falling into a local minimum. This situation did not occur after the optimization of the intelligent algorithm, and the maximum error values were all reduced. The GA-BPNN model predicted BW error values of between 0.5255 and 4.1833% and BH error values of between 1.2062 and 5.7697%. The PSO-BPNN model predicted BW error values of between 0.3957 and 2.5122% and BH error values of between 0.126 and 4.9102%. The BAS-BPNN model predicted BW error values of between 0.16615 and 4.80278% and BH error values of between 0.3119 and 4.7162%. From the above, the error values could be reduced by the optimization of the GA and PSO and BAS, respectively, but the optimization of the BAS and PSO was better.

To further comprehensively and quantitatively assess the models’ prediction performance and generalization capacity, three performances—the mean absolute error (MAE), mean absolute percentage error (MAPE), and root mean square error (RMSE)—were chosen as assessment standards for model accuracy [37]:(7)$$MAE=\frac{1}{n}\sum_{i=1}^{n}\left|y_{i}^{\mathrm{expr}}-y_{i}^{pre}\right|$$
(8)$$MAPE=\frac{1}{n}\sum_{i=1}^{n}\left|\frac{y_{i}^{pre}-y_{i}^{\mathrm{expr}}}{y_{i}^{\mathrm{expr}}}\right|\times 100\%$$
(9)$$RMSE=\sqrt{\frac{1}{n}\sum_{i=1}^{n}{\left(y_{i}^{\mathrm{expr}}-y_{i}^{pre}\right)}^{2}}$$
where $y_{i}^{pre}$ and $y_{i}^{\mathrm{expr}}$ are, respectively, the predicted and experimental data (BW and BH) concerning the additive manufactured component and *n* is the total sample number used in the test set.

As shown in Figure 10, the MAE, MAPE, and RMSE of the GA-BPNN, PSO-BPNN, and BAS-BPNN models in predicting the BW and BH of the additive manufactured component were significantly smaller than those of the single BPNN model, indicating that the prediction performance of these three models was significantly better than that of the BPNN model. In predicting the BH, the three evaluation indicators of the BAS-BPNN model were 0.0710, 2.2808, and 0.0856, each of which was the smallest, and the variability of the indicators was the smallest relative to the GA-BPNN and PSO-BPNN models. In terms of predicting the BW, the MAE and MAPE of the BAS-BPNN model were smaller at 0.1191 and 1.4683, while the RMSE of the PSO-BPNN model was smaller at 0.1410. Considering that the model was synchronized to predict the BW and BH, the BAS-BPNN model had a better prediction capacity.

Combined with the geometries predicted by the BAS-BPNN model, we evaluated the influence of process parameters on the surface quality of the deposited layers. The deposited layers were continuous at each process parameter, but the parameters had a significant effect on the surface morphology. Moreover, four main characteristics of the deposited layer, including being well-formed, too narrow, too thick, and slightly wavy, were observed. Based on this, we determined the process window regarding the surface-forming quality of the SS316L fabricated by CMT-WAAM, as shown in Figure 11.

Black spots: Well-formed. The deposited layer is continuous with a smooth surface and uniform size distribution.

Orange spots: Too narrow. The narrow shape may be caused by insufficient energy. At a low WFS (3–5 m·min^−1^) and TS (10–14 mm·s^−1^), in this CMT process, the WFS was small, the overall input power was small, the TS was too large, and the energy obtained by the system was only able to melt a small amount of wire and matrix materials, resulting in a relatively small molten pool, and the solidification process of the liquid metal was completed in a relatively short time.

Blue spots: Too thick. The deposition layers have a greater width and height. The reason is that at a higher WFS and a lower TS, a larger molten pool and sedimentary layer will be produced.

Green spots: Slightly wavy. There is a wavy morphology in the deposition layer. It was determined that the TS was high, the welding torch moving speed was too large, the arc combustion was unstable, the system heat input was relatively small, and the weld toe on both sides of the track was not completely melted.

The influence of process parameters on the feature geometry of single-layer parts is shown in Figure 12. The width and height of the feature geometric parameters tended to decrease with increases in the TS. This is because when the system input energy is constant, the TS increases, the line energy density decreases, and the amount of metal filled per unit time decreases. As the WFS increases, the width and height increase. This is because the higher the WFS, the greater the input energy, and the more SS316L wire will melt per unit time; the substrate material will also melt, so that in the process of a melting pool flow, the higher the level of metal filling in the deposit layer.

### 4.2. Thermal Evolution

Based on the dimension predicted by the BAS-BPNN model, the heat transfer behavior of SS316L under different WAAM processes (WFS = 8 m·min^−1^, TS = 10, 12, 14 mm·s^−1^) was studied. To verify the validity of the model, we compared experimental and simulation results for melt pool sizes under different parameters. As shown in Figure 13, when the TS was 10 mm·s^−1^, 12 mm·s^−1^, and 14 mm·s^−1^, the errors of the melt pool width were 1.36%, 3.09%, and 3.58%; the errors of the melt pool depth were 0.14%, 1.08%, and 1.35%. It can be seen that the prediction error was very small, which shows that the combined model proposed in this paper can perform the calculation of the temperature field well.

In order to analyze the effect of CMT process parameters on temperature and the microstructure more intuitively, the line energy density (LED) was quoted as the reference index:(10)LED=UI/TS

The temperature results under the three processes are shown in Figure 14. The highest temperature of the deposited layer was, respectively, 2368 °C, 3695 °C, and 2818 °C when the TS was 14 mm·s^−1^, 12 mm·s^−1^, and 10 mm·s^−1^. The temperatures far exceeded the liquid phase temperature of the SS316L (1450 °C) and were able to melt and metallurgically bond 316L (see Figure 14a). Moreover, the temperature in the molten pool increased as the TS decreased. This is because the decrease in TS leads to an increase in LED and a longer arc action time, so the molten pool center temperature increases.

### 4.3. Microstructure Characteristic

The grain structure is closely related to several properties in the WAAM process. The microstructure of different regions of the deposited layer cross-section is shown in Figure 15, where there were significant differences in the microstructure of deposited metals in different regions. Figure 15a shows the morphology of the cellular crystal structure on the bottom part. These crystals mainly exhibited a cellular structure with well-developed primary branching and almost indistinguishable secondary branching. As solidification proceeded, the grain structure transformed into columnar dendrites (see Figure 15b), and an equiaxed dendrite structure appeared in the surface region near the deposition layer top zone (see Figure 15c). The evolution of this grain structure is related to the heat transfer behavior in the melt region, where the solidification parameters (G and R) are determinants of crystal growth [38]:(11)$$G=\left\|\nabla T\right\|=\left\|\frac{\partial T}{\partial x}i+\frac{\partial T}{\partial y}j+\frac{\partial T}{\partial z}k\right\|$$
(12)R=v⋅i⋅n=v⋅cosα

The variation in the G/R and G×R (cooling rate) all the way up from the melt zone’s bottom is displayed in Figure 15d. Based on the G/R results and the judgment conditions [39,40], it can be determined that interface destabilization occurred as well as constitutional subcooling during the solidification process. At the early stage of solidification, the temperature gradient was relatively large, the solidification rate was small, the solid–liquid flat interface was destabilized, the interface was raised in certain places, the solute aggregated, the composition subcooling zone was small, the raised part did not have a large extension, and the cellular microstructure was formed. As solidification proceeded, the raised part of the S/L interface continued to extend into the subcooled liquid phase region, while branching occurred laterally, forming the columnar dendrites. At the end of solidification, the G was smaller, and the R increased, and the G/R became smaller, which led to independent nucleation and growth at the top of the melt region, which inhibited the growth of columnar dendrites and promoted the transformation of columnar dendrites to equiaxed crystals (CET), and the grain microstructure in the top surface region was equiaxed. It can be seen that there was such a solidification mechanism in the accretion process, where a higher G/R induced the growth of columnar dendrites and a relatively lower G/R tended to cause equiaxial growth. The G/R critical value for the grain morphology transformation could be explored more deeply in the future.

Since the main grain structure in the deposited layer is columnar dendrites, the effect of the TS on the columnar dendrites of SS316L was investigated, and the grain structure is shown in Figure 16. When the TS was 14 mm·s^−1^, 12 mm·s^−1^, and 10 mm·s^−1^, the average primary dendrite spacing in the equiproportional region was 12.74 μm, 15.21 μm, and 17.37 μm, respectively. It can be seen that the primary dendrite spacing became larger as the TS decreased. This law is related to the G×R during the solidification process of the molten pool. Figure 17 shows the G×R of feature points in each region shown in Figure 16. When the TS increased, the G×R in the middle of the deposited layer increased significantly during solidification, resulting in the formation of finer columnar dendritic structures after solidification.

Figure 18 shows the XRD pattern of the SS316L; the three highest overall strength austenite peaks in the deposited layer were *γ* (111), *γ* (200), and *γ* (220). Two ferrite peaks *δ* (110) and *δ* (200) were also observed, indicating that the clad layer was predominantly made of austenite and a small amount of *δ*-ferrite. This finding is in line with the microstructure shown in Figure 15 and Figure 16, which shows that the layer was mostly constituted of austenite. Moreover, the diffraction peak area of the *δ*-ferrite of the specimen was larger when the TS was 14 mm·s^−1^, which is presumed to be due a higher *δ*-ferrite content. This is directly related to its solidification behavior. Figure 19 displays the elemental weight results obtained through quantitative EDS analysis. $Cr_{eq}/Ni_{eq}$ can be calculated to be greater than 1.5 based on the elements’ weights ($Cr_{eq}=Cr+Co+0.7Nb$, $Ni_{eq}=Ni+35C+20N+0.25Cu$), and the sedimentary layer solidified in FA mode (L→L+δ→L+δ+γ→δ+γ→γ) [41,42]. The cooling rate (G×R) was larger at TS = 14 $mm\cdot s^{-1}$ (Figure 17), and the rate of heat dissipation during solidification was accelerated, leading to a shorter time required for a eutectic reaction and solid-phase transformation. As a result, the amount of *δ*-ferrite precipitated from the liquid phase was higher.

## 5. Conclusions

In this study, A model combining ANN and FE was established. Based on the intelligent algorithm and BPNN model, the prediction of the width and height of a SS316L deposited layer under different process parameters was realized. Then, FE temperature simulation based on the predicted geometry was implemented, and the intrinsic connection between solidification parameters, the microstructure, and the microhardness was investigated by the temperature field. The main conclusions are as follows:(1)The predicted width and height of the deposited layer under different processes were in good agreement with the experimental data. The errors of the width and height predicted by the models (GA-BPNN, PSO-BPNN, BAS-BPNN model) were all less than 6%. Besides this, the MAE, MAPE, and RMSE of the BAS-BPNN model were always smaller compared to other models, which means that the BAS-BPNN model had a better prediction capacity in the geometry dimension.(2)Process windows were established based on predictions and experiments. Continuous, stable, good melt tracks could be formed over a wide range of parameters (WFS (3–4 m·min^−1^) and TS (3–6 $mm\cdot s^{-1}$); WFS (5 m·min^−1^) and TS (4–10 mm·s^−1^); WFS (6–7 m·min^−1^) and TS (5–14 mm·s^−1^); WFS (8 m·min^−1^) and TS (6–14 mm·s^−1^)). The width and height of the single track showed a decreasing trend when the TS was increased and an increasing trend when the WFS was decreased.(3)The melt pool obtained from the temperature simulation agreed well with the experimental results, and the coupled model was able to simulate effectively. When the TS was 14 mm·s^−1^, 12 mm·s^−1^, and 10 mm·s^−1^, the molten pool width errors were 1.36%, 3.09%, and 3.58%, and the molten pool depth errors were 0.14%, 1.08%, and 1.35%, respectively. The highest temperature in the molten pool increased as the TS decreased.(4)The microstructural evolution during rapid solidification in the SS316L WAAM was related to its thermal behaviour. Decreases in G/R induced a change in the crystal structure from columnar dendritic crystals to equiaxed dendritic crystals. Due to the increase in cooling rate, the primary dendrite spacing became larger and the δ-ferrite content increased.

## Figures and Tables

**Figure 1 micromachines-15-00615-f001:**
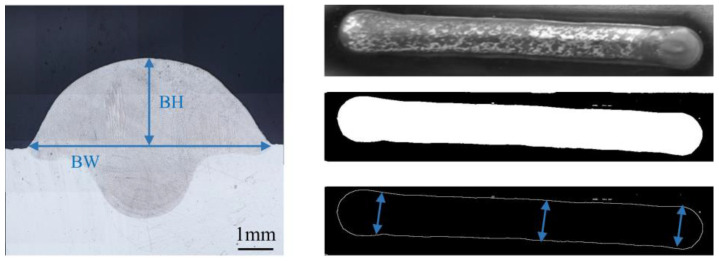
Single-track deposition layer geometry morphology.

**Figure 2 micromachines-15-00615-f002:**
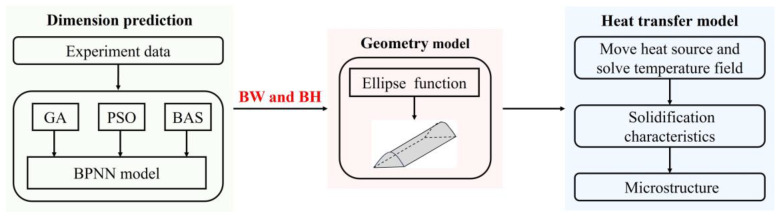
The computational flow chart of the dimension prediction–heat transfer coupling model.

**Figure 3 micromachines-15-00615-f003:**
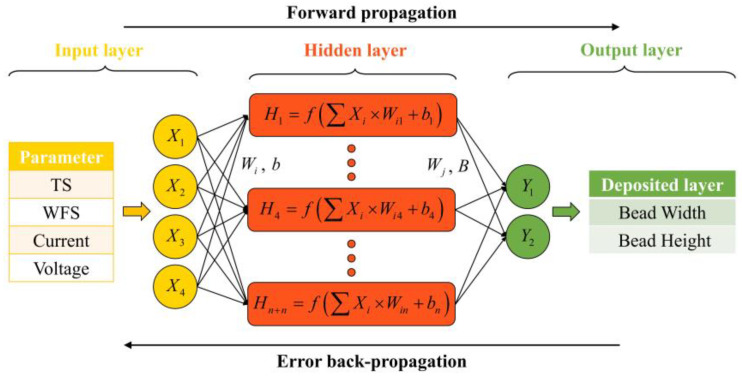
BPNN structure.

**Figure 4 micromachines-15-00615-f004:**
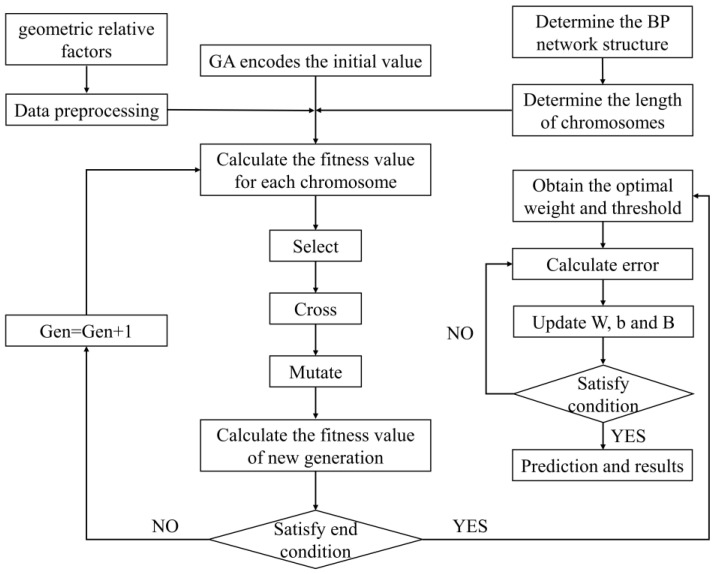
GA-BPNN flow chart.

**Figure 5 micromachines-15-00615-f005:**
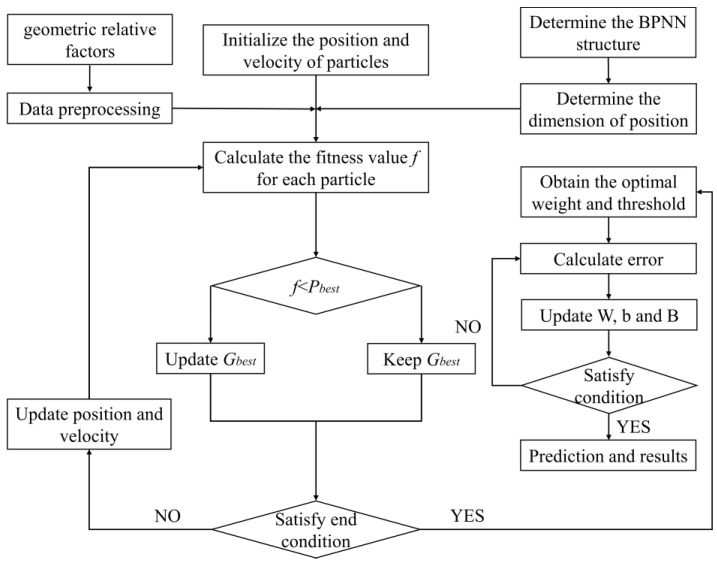
PSO-BPNN flow chart.

**Figure 6 micromachines-15-00615-f006:**
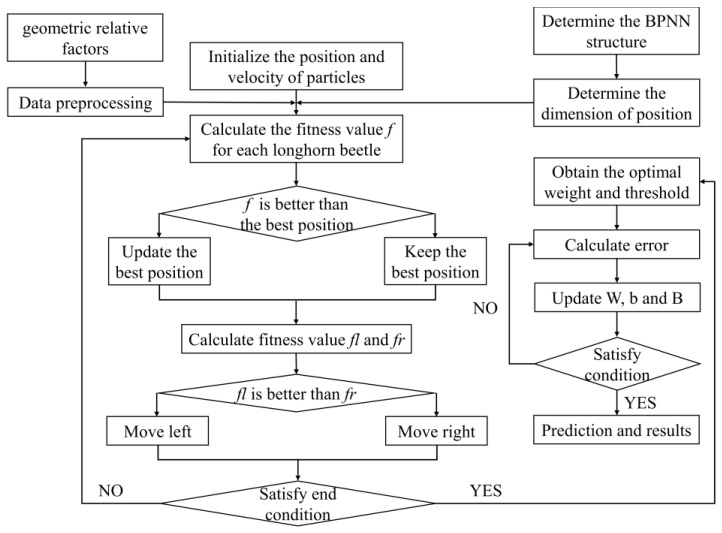
BAS-BPNN flow chart.

**Figure 7 micromachines-15-00615-f007:**
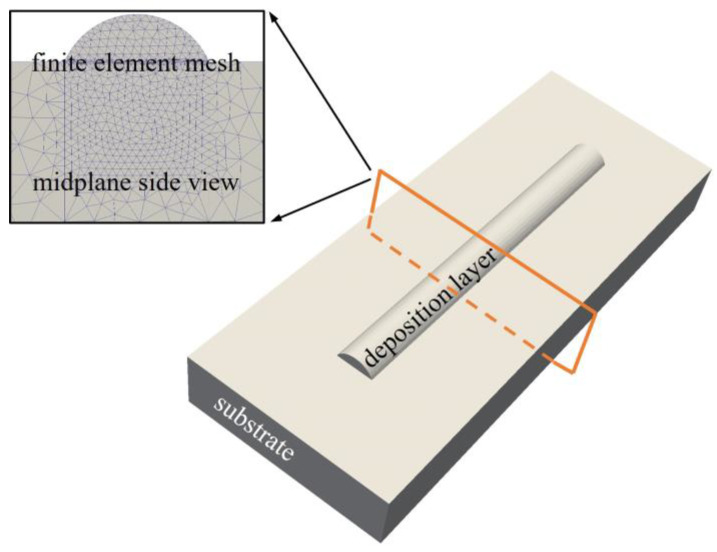
Finite element geometric model.

**Figure 8 micromachines-15-00615-f008:**
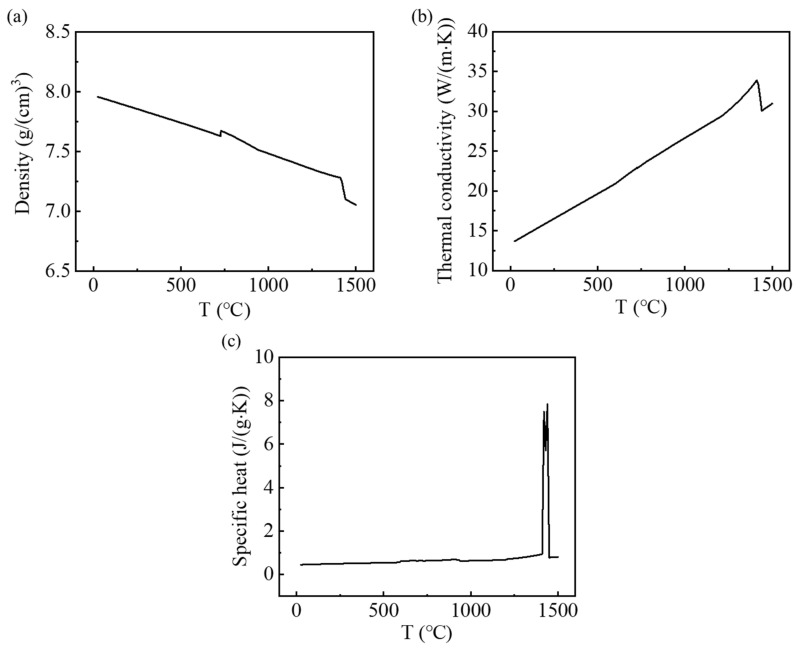
The SS316L’s physical properties: (**a**) density; (**b**) thermal conductivity; (**c**) specific heat.

**Figure 9 micromachines-15-00615-f009:**
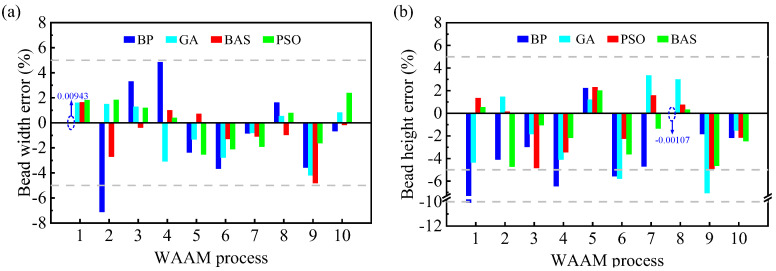
Prediction error for geometric dimensions: (**a**) BW; (**b**) BH.

**Figure 10 micromachines-15-00615-f010:**
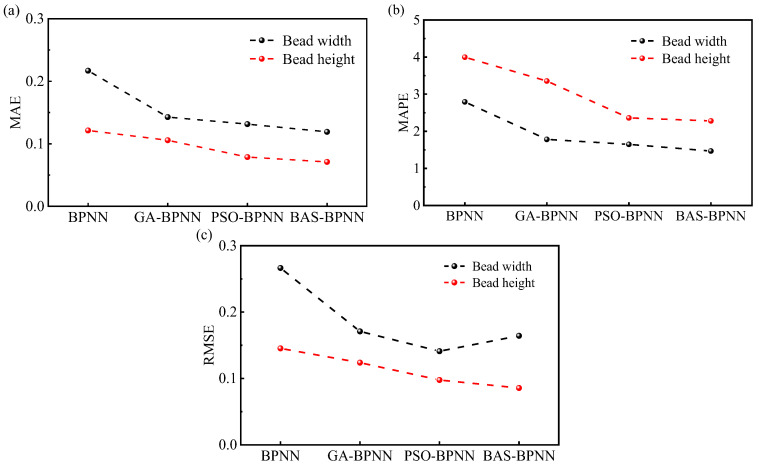
The evaluation indicators for the models: (**a**) MAE; (**b**) MAPE; (**c**) RMSE.

**Figure 11 micromachines-15-00615-f011:**
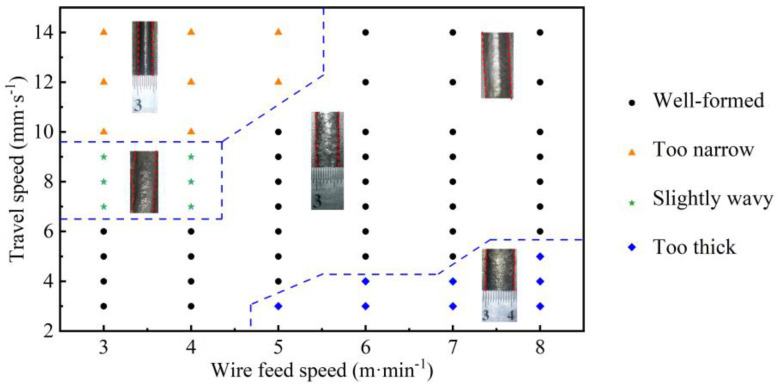
The process window for single-track morphology deposited by WAAM.

**Figure 12 micromachines-15-00615-f012:**
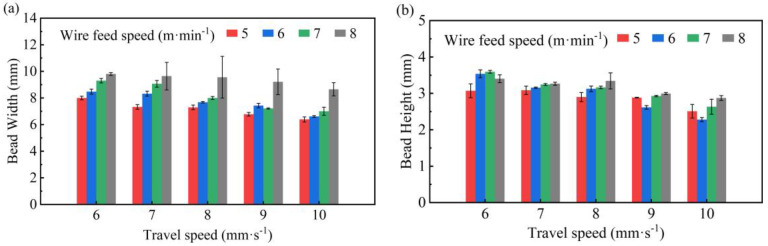
Effect of TS and WFS on the geometry dimensions: (**a**) bead width; (**b**) bead height.

**Figure 13 micromachines-15-00615-f013:**
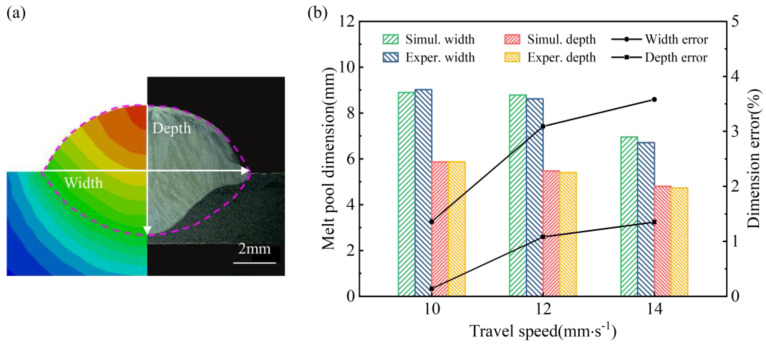
Model validation: (**a**) molten pool cross section; (**b**) molten pool dimension.

**Figure 14 micromachines-15-00615-f014:**
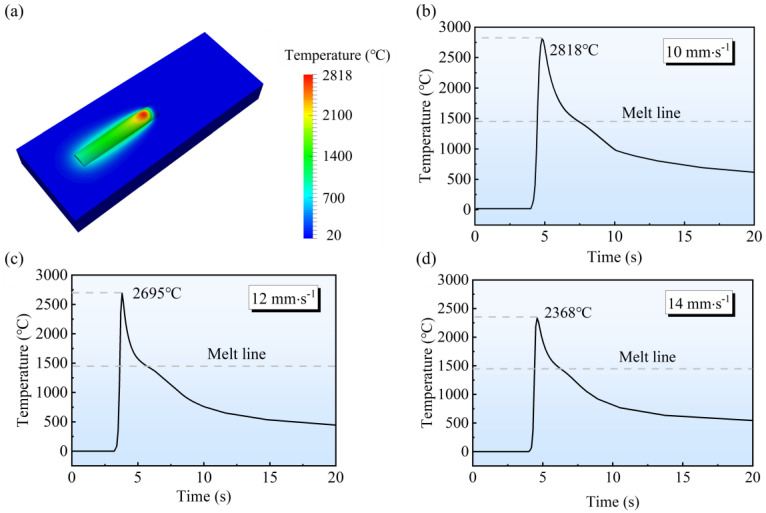
Temperature under different TSs: (**a**) temperature distribution; (**b**) 10 mm·s^−1^; (**c**) 12 mm·s^−1^; (**d**) 14 mm·s^−1^.

**Figure 15 micromachines-15-00615-f015:**
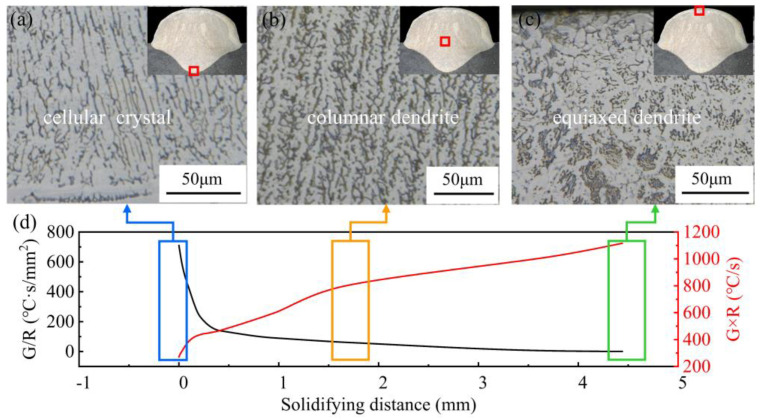
Solidification parameters and microstructure at TS = 14 mm·s^−1^, WFS = 8 m·min^−1^: (**a**) microstructure at the bottom; (**b**) microstructure at the middle; (**c**) microstructure at the top; (**d**) G×R and G/R.

**Figure 16 micromachines-15-00615-f016:**
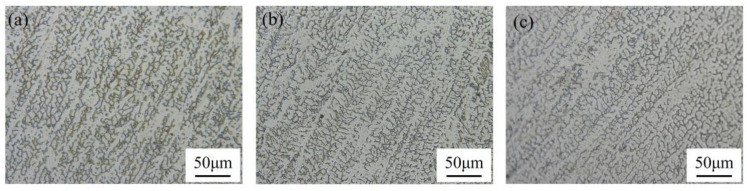
Microstructure of the middle zone of the monorail at equal proportions under different TSs: (**a**) 14 mm·s^−1^; (**b**) 12 mm·s^−1^; (**c**) 10 mm·s^−1^.

**Figure 17 micromachines-15-00615-f017:**
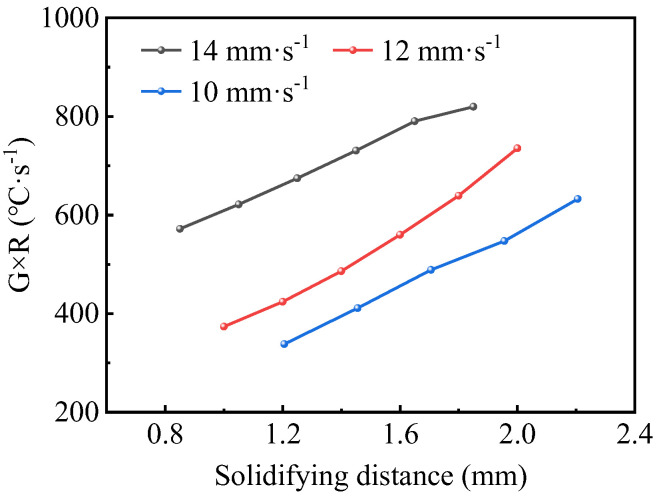
*G* × *R* under different TSs.

**Figure 18 micromachines-15-00615-f018:**
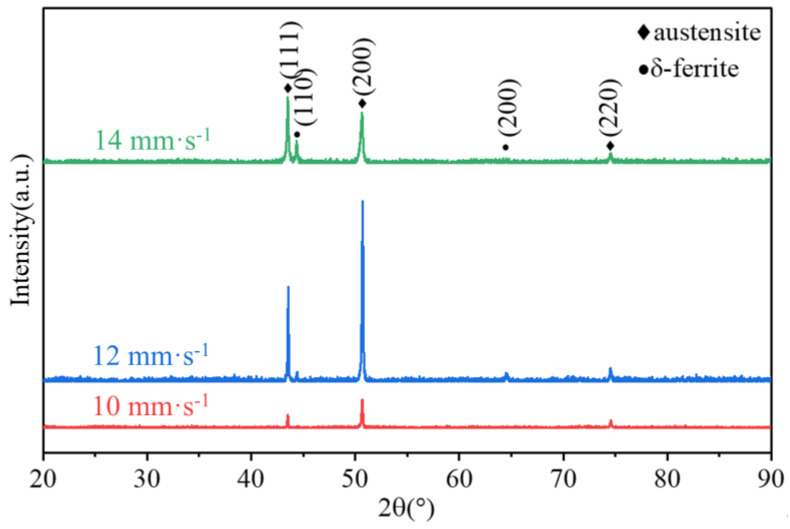
XRD diffraction patterns of the WAAM-ed parts under different TSs.

**Figure 19 micromachines-15-00615-f019:**
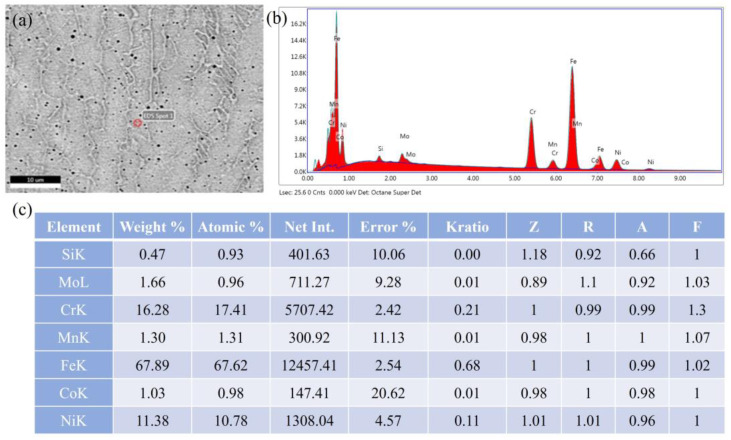
EDS quantitative analysis result: (**a**) calibration region; (**b**) EDS spectrum; (**c**) element weight.

**Table 1 micromachines-15-00615-t001:** Chemical composition of the SS316L wire (wt.%).

Element	Cr	Ni	Mo	Mn	Si	C	S	P	N	Fe
Content	18.39	12.5	2.25	1.69	0.81	0.02	0.015	0.015	0.013	balance

**Table 2 micromachines-15-00615-t002:** The WAAM process parameters.

Parameter	Units	Value
Transient voltage (TU)	V	14.4–24.5
Transient current (TI)	A	122–243
Travel speed (TS)	mm·s^−1^	3–14
Wire feed speed (WFS)	m·min^−1^	3–8.3
Gas flow rate	L·min^−1^	20

**Table 3 micromachines-15-00615-t003:** The predicted and experimental values of the BW.

WAAM Parameter			Value (mm)				
Num.	WFS	TS	BPNN	GA-BPNN	PSO-BPNN	BAS-BPNN	Exper.
1	8.3	9	8.9909	9.1334	9.1522	9.1359	8.99
2	8	12	6.8121	7.4419	7.4676	7.1362	7.3333
3	6	6	8.8909	8.7163	8.7083	8.5752	8.6067
4	7	8	8.3067	7.6802	7.9547	8.0022	7.9233
5	4	5	7.4564	7.5365	7.4448	7.6908	7.6367
6	3	6	6.2372	6.2942	6.3382	6.3904	6.4733
7	8	10	8.1115	8.1146	8.0252	8.091	8.18
8	5.5	7	7.8946	7.8108	7.8308	7.6952	7.77
9	5	5	8.4923	8.4383	8.6645	8.3837	8.8067
10	4.5	4	8.3912	8.5153	8.6473	6.193	8.4467

**Table 4 micromachines-15-00615-t004:** The predicted and experimental values of the BH.

WAAM Parameter			Value (mm)				
Num.	WFS	TS	BPNN	GA-BPNN	PSO-BPNN	BAS-BPNN	Exper.
1	8.3	9	2.7372	2.9113	3.084	3.0598	3.0433
2	8	12	2.4654	2.6072	2.5732	2.4488	2.57
3	6	6	3.5065	3.5476	3.439	3.5756	3.6133
4	7	8	2.9845	3.0601	3.0807	3.1213	3.19
5	4	5	3.1987	3.1669	3.2012	3.192	3.1292
6	3	6	2.3171	2.3118	2.3983	2.3647	2.4533
7	8	10	2.7386	2.9693	2.9183	2.8347	2.8733
8	5.5	7	3.137	3.2306	3.16	3.1468	3.137
9	5	5	3.6982	3.5011	3.5817	3.5923	3.7667
10	4.5	4	3.669	3.693	3.6699	3.6582	3.75

## Data Availability

Data are contained within the article.

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
