# Peer review of "Dimension Prediction and Microstructure Study of Wire Arc Additive Manufactured 316L Stainless Steel Based on Artificial Neural Network and Finite Element Simulation"

_micromachines, 2024, doi:10.3390/mi15050615_

Round 1
Reviewer 1 Report
Comments and Suggestions for Authors
This paper used artificial neural networks and finite element simulation to predict the dimension accuracy and the microstructure of wire arc additive 2 manufactured 316l stainless steel. The manuscript was read with interest by the reviewer. It is well written and presents useful information. I recommend to accept it for publication with minor revision. The comments are below:
(1) In the introduction section, the numerical computation methods should be classified as artificial neural networks, finite element simulation, and others.
(2) The computation process should be described more detailed, making it repeatable.
(3) More experimental data should be added into the text part. They are really useful information for the colleagues.
(4) In Fig. 19, the serial number (b and c) overlap the figures. It needs to be revised.
(5) The microstructure prediction in this study is very weak. I suggest to change the title of the manuscript to make it more suitable.
Comments on the Quality of English LanguageMinor editing of English language required
Reviewer 2 Report
Comments and Suggestions for Authors
